# Effects of Germinated Lentil Flour on Dough Rheological Behavior and Bread Quality

**DOI:** 10.3390/foods11192982

**Published:** 2022-09-23

**Authors:** Denisa Atudorei, Silvia Mironeasa, Georgiana Gabriela Codină

**Affiliations:** Faculty of Food Engineering, Stefan cel Mare University of Suceava, 720229 Suceava, Romania

**Keywords:** germinated lentil flour, refined wheat flour, fundamental rheology, dough microstructure, bread quality

## Abstract

The present study analyzed the effects of germinated lentil flour (LGF) addition at different levels in wheat flour (2.5%, 5%, 7.5%, and 10%), on dough rheological behavior, dough microstructure, and bread quality. Creep-recovery tests showed that the dough samples with high levels of LGF addition presented a higher resistance to flow deformability of the dough. Dough microstructure as analyzed using EFLM showed an increase in the protein area (red color) and a decrease in the starch (green color) amount with the increased level of LGF addition in the wheat flour. It was found that the LGF addition led to the improvement of the porosity, specific volume, and elasticity of the bread samples. The breads with LGF addition were darker and had a slightly reddish and yellowish tint. The bread textural parameters highlighted significant (*p* < 0.05) higher values for firmness and gumminess and significant (*p* < 0.05) lower ones for cohesiveness and resilience for the bread with LGF addition when compared with the control. The bread samples with a 2.5% and 5% addition had a more dense structure of the crumb pores. Regarding sensory evaluation, the bread samples with LGF addition in the wheat flour were well appreciated by the consumers. The addition also was desirable due to the fact that it supplemented bread with a greater amount of protein and minerals due to the composition of lentil grains. Therefore, LGF could be successfully used as an ingredient for bread making in order to obtain bread with an improved quality.

## 1. Introduction

Currently, consumers demand an increasing amount of healthy food products [1]. The concept of healthy food mainly comprises two converging aspects: one is related to a reduction in the amount of chemical additives used for food production and the other refers to a balanced nutritional profile of food products that meet the daily requirements. These aspects influence the trend related to the production of new kinds of food or to the improvement of existing ones [2]. In modern times, special attention is paid to the diversification of product ranges to obtain new products [3]. Consumers are always looking for something new that can satisfy them in terms of quality and nutrition but that does not require extremely high economical costs.

Worldwide, the most consumed food product in various forms is bread, for which the base raw material is wheat flour [4]. Although it is a kind of food situated at the base of the food pyramid that is consumed by all age groups, the most consumed is that made from white wheat flour, for which the nutritional value is reduced due to the wheat flour refining process [5]. Moreover, in order to obtain bread products of high quality, a large amount of chemical additives are used in bread recipes. In order to nutritionally improve the quality of white wheat bread, different types of legumes may be used that can balance the nutritional value of the bread [6] and reduce the effect of refining flour, a process in which a number of nutrients are lost [7,8]. Their use in food products is recommended after some processing techniques in order to minimize their unpleasant flavor and antinutritional contents. Such a technique is known as germination, which reduces the antinutritional compounds from legume grains and improves the nutritional content [9]. Some of the beneficial effects of the germination process on grains include an increase in the bioavailability of nutrients, a reduction in antinutrient factors, and an increase in the amount of nutritional compounds that are released during germination that were previously bound to other compounds. [10,11]. Furthermore, it has been pointed out that during germination, the activation of specific enzymes in the grains takes place; these enzymes can successfully replace some chemical additives that may be exogenously used in wheat flour in order to improve the bread-making process and the quality of bakery products. Therefore, the addition of flour from various germinated legumes in a bread recipe could be a viable alternative to the idea of obtaining bread with superior nutritional properties but also with an improved quality. The results of scientific works in the field showed that the germination process also induces a series of changes in the grain structure: it modifies the crystallinity of starch and influences viscosity and retrogradation enthalpy. All these suggest that the germination process can also modify the structure of the food products in it is incorporated [12].

Lentils have an important place among vegetables. They are also an important source of protein in vegetarian diets [13]. According to FAOSTAT (2020) [14], lentil consumption has increased in recent years, probably due to its positive effects on human health. Wilson et al. [15] concluded that lentil consumption improved insulin sensitivity in overweight or obese people due to the fact that it promoted a lower blood glucose level and improved the body’s response to insulin. Zahradka et al. [16] reported that the consumption of lentils improved arterial elasticity, countered atherosclerotic disease, and led to a lower blood pressure. Papandreou et al. [17] reported that the consumption of lentils decreased the possibility that people would face obesity, cancer, chronic diseases, and hypertension, during their life. Lentils have a balanced nutritional profile. The nutritional content of lentil flour consists of 22.7% proteins (containing all essential amino acids), 51% carbohydrates, 13.8% dietary fiber, and 1% lipids [18]. Due to its nutritional value, the possibility of using LGF as an ingredient in bread-making recipes is of scientific interest. Moreover, the use of lentils in germinated form is indicated due to the benefits brought by the germination process in its nutritional profile. In a previous study, it was stated that exposing red lentils (*Lens culinaris Merr*.) to the germination process for four days led to an increase in the amount of proteins and ash and a decrease in the amount of lipids and carbohydrates. Moreover, it was observed that due to germination, the amount of sodium, magnesium, iron, and zinc in the lentil grains increased [19].

While different studies have reported on the possibility of using lentil flour as an ingredient in the bread-making process [20], very few have explored the possibility of using lentil flour in a germinated form in bread recipes [10]. Ungureanu-Iuga et al. [21] reported that the addition of LGF in wheat flour changed dough rheological behavior significantly, leading to a decrease in the dough extensibility, stability, and baking strength and an increase in the dough consistency and viscoelastic moduli. In a study by Razavi et al. [22] the addition of LGF to Sangak bread led to an increase in the dough development time and bread hardness. From the sensory point of view, the best appreciation was received for the bread with 5% LGF addition in the recipe. Hernandez-Aguilar et al. [23] reported that LGF addition in bread making led to a decrease in the bread cohesiveness and an increase in its hardness. Marti et al., 2017 [24] highlighted the fact that the germination process could be used successfully for obtaining bakery products. They studied the advantages of using refined flour from sprouted wheat and concluded that this led to an increase in gas production during the proofing process, which had the effect of increasing the specific volume of the bread. At the same time, the obtained bread had an improved crumb softness during storage. These results were mainly due to the enzyme activity from the sprouted flour that was used in the bread making.

The present study analyzed the effect of LGF addition in wheat flour on dough rheological behavior, its microstructure, and bread quality in terms of the physicochemical, textural, structural, and sensory characteristics. Thus, by highlighting the optimal level of LGF addition, it was possible to quantify its positive effect on the quality of the bread and, implicitly, its technological role in bread making.

## 2. Materials and Methods

### 2.1. Materials

The main raw material used for the bread-making process was soft wheat flour (type 650). This was purchased from a bakery factory (S.C. Dizing S.R.L. company from Brusturi, Neamt, Romania). Germinated lentil flour (LGF) was obtained from *Lens culinaris Merr. lentils*. For the germination of lentils, the following parameters were followed: a temperature of 25 °C, a constant humidity of 80%, and a time of germination of four days. Germination was performed in exclusive dark conditions according to the procedure described in previous study [19]. After the germination process was completed, the lentils were lyophilized to reduce their moisture in order to be ground. For lyophilization, a Biobase BK-FD12 lyophilizer (Jinan, China) was used. Lyophilization (freeze-drying) was performed at −50 °C for 24 h at a pressure of 10 Pa. Germinated lentils was ground by using an LM 3310 laboratory mill (Perten Instruments, Hägersten, Sweden). In order to preserve the germinated lentil flour, this was vacuum-packed in polythene bags and stored at room temperature for a maximum of six months.

The wheat flour and LGF were analyzed in accordance with ICC standard methods [25] as follows: ash content (ICC 104/1), fat content (ICC 136), protein content (ICC 105/2), and moisture content (ICC 110/1). In the case of wheat flour, the wet gluten content (ISO 21415-2:2015), the gluten deformation index (Romanian standard SR 90/2007), and the falling number value (ICC 107/1 method) were also determined.

### 2.2. Dough Fundamental Rheological Properties

The dough rheological properties in the case of all samples were analyzed by performing creep and recovery tests at 25 °C using a HAAKE MARS 40 rheometer (Thermo-HAAKE, Karlsruhe, Germany). For these tests, a nonserrated parallel plate geometry with a diameter of 40 mm and a gap width of 2 mm was used. The samples were mixed using an Alveo Consistograph while taking into account the optimum Consistograph water absorption. After preparing the samples, they were placed in the rheometer plate. The samples were rested for 5 min before analysis for relaxation and for temperature stabilization. The determinations were made at constant shear stress of 50 Pa following a frequency sweep from 1 to 20 Hz in the range of linear viscoelasticity. The times for creep and recovery stage were 60 s and 120 s, respectively. The equations that describe the specific parameters for these determinations were previously described in [26].

### 2.3. Dough Microstructure

The dough samples with or without LGF addition were analyzed from the microstructural point of view using a Motic AE 31 (Motic, Optic Industrial Group, Xiamen, China) equipped with LWD PH 203 catadioptric objectives (N.A. 0.4). The preparation of the samples and the epifluorescence light microscopy (EFLM) analysis were performed according to the method described in our previous study [26]. Thus, according to this methodology, a thin portion was cut from the dough samples and immersed in a fixing solution for at least 1 h. This fixing solution was composed of 1% rhodamine B and 0.5% fluorescein (FITC) in 2-methoxyethanol. The substances were supplied by Sigma-Aldrich, Steinheim am Albuch, Germany. Fluorescein and rhodamine B were used as specific fluorescent dyes to detect proteins (rhodamine B) and starch (fluorescein) in the dough samples. After immersing the dough in the fixing solution, EFLM images were obtained and then analyzed using ImageJ (v. 1.45, National Institutes of Health, Bethesda, MD, USA) software.

### 2.4. Bread Making

The ingredients used to prepare the bread samples were: white wheat flour (type 650), LGF (at variable levels of 2.5%, 5%, 7.5%, and 10%), 1.5% sodium chloride, 3% compressed yeast of *Saccharomyces cerevisiae,* and water according to the water absorption capacity value of wheat-germinated lentil flours. LGF replaced white wheat flour in various proportions that were previously specified. The amount of water used was determined according to the values of water absorption that were determined using the Alveo Consistograph. The water additions were as follows: 54.3% for the control sample and 53.7%, 53.1%, 52.6%, and 52.2% for the samples with 2.5%, 5%, 7.5%, and 10% LGF addition, respectively, in the wheat flour. To prepare the bread samples, the following steps were followed: dosing of the ingredients, mixing, dividing the dough obtained, leavening the dough, and baking the samples. A heavy-duty mixer (KitchenAid, Whirlpool Corporation, Benton Harbor, MI, USA) was used to mix the ingredients. The ingredients were mixed for 15 min. After kneading, the dough obtained was divided into three parts of 400 g each. The fermentation of the samples resulting from the modeling step was done in a fermentation chamber (PL2008, Piron, Italy) at a temperature of 30 °C for 60 min at 85% relative humidity. The samples were baked in an electrical bakery convection oven with steam production, ventilation, and humidification (PF8004 D, Piron, Italy) at a temperature of 220 °C for 30 min using the steam system of the oven in the first 2 min of baking.

### 2.5. Bread Quality Evaluation

#### 2.5.1. Bread Physical Characteristics

The physical characteristics of the bread samples without and with LGF addition were analyzed by determining the values for the specific volume, porosity, and elasticity of the samples. The rapeseed displacement method (AACC Method 10–05.01) [27] was used to determine the values for the specific volume. According to this method, rapeseeds were first placed in a container used to measure the specific volume of the bread samples. Then, the volume of the rapeseeds was measured using a graduated cylinder. After this step, the bread samples were placed one by one in the same container and the rapeseeds were poured on top of them until the container was filled and the bread was covered. After that, the volume of rapeseeds used was measured. The specific volume of the samples was then calculated using the equation presented in a study by Feili et al., 2013 [28]. The SR 91:2007 standard method was taken into account in the determination of the values of porosity and elasticity. The values of the crumb elasticity of the bread samples were determined by considering the hardness of the crumb when a uniaxial compressive force was applied. The porosity of the bread samples was evaluated by taking into account the total volume of hollows from a known volume of bread crumb [29,30].

#### 2.5.2. Color Parameters

The color characteristics (for crumb and crust of the bread samples) were determined using a Konica Minolta CR-400 colorimeter (Tokyo, Japan). With the help of this colorimeter, the values for the following parameters were measured: *L** (darkness/brightness), *b** (shade of blue/yellow), and *a** (shade of red/green). These determinations were based on the CIE Lab* color system and were made using UV–vis technology. For these analyses, before starting the determinations, the colorimeter was calibrated by scanning the standard white surface calibration plate (*L** = 97.63, *a** = 0.01, and *b** = 1.64), which used the standard illuminant D_65_ (working in daylight) and a 0° viewing angle.

#### 2.5.3. Texture Profile Analysis

A TVT-6700 texturometer device (Perten Instruments, Hägersten, Sweden) was used to determine the textural characteristics of the bread samples. For the determination, the texturometer used was equipped with a 10 kg load cell. Slices of bread of that were 50 mm tall were used for the determination. These were subjected to two compression cycles to up to 20% of their initial height. The analysis was conducted using a 45 mm cylindrical probe. The protocol used was: a trigger force of 5 g, a speed of 1.0 mm/s, and a recovery period between compressions of 15 s. The textural parameters that were measured in this test were: firmness, gumminess, cohesiveness, and resilience.

#### 2.5.4. Crumb Structure

To analyze how the addition of LGF influenced the crumb structure of the bread samples, a Motic SMZ-140 stereo microscope (Motic, Xiamen, China) was used at a resolution of 2048 × 1536 pixels with a 20× objective.

#### 2.5.5. Sensory Analysis

The characteristics that were taken into account in the sensory analysis were: appearance, color, aroma, taste, smell, texture, and global acceptability. The sensory analysis of the bread samples was conducted with the help of a panel of 20 semitrained judges using a 9-point hedonic scale from 1 to 9 in which 1 = dislike extremely, 5 = neither like nor dislike, and 9 = like extremely. At the end, the arithmetic average was calculated for the points awarded and the average score for each judge was established.

### 2.6. Statistical Analysis

Statistical processing of data obtained from specific determinations for all bread samples was done using the Statistical Package for Social Science (v.16, SPSS, Chicago, IL, USA). Data were expressed as the mean ± standard deviation and a one-way analysis of variance (ANOVA) was used with a Tukey’s test at a level of 5% of significant differences.

## 3. Results

### 3.1. Flour Characteristics

The wheat flour presented the following characteristics: 0.66% ash content, 14.6% moisture, 1.12% fat, 12.3% protein, 30.4% wet gluten, and a 3 mm gluten deformation index. The value of the falling number index was 356 s. According to these data, the white wheat flour had a low α-amylase activity and was one of strong quality for bread making [31]. LGF presented the following characteristics: 8.8% moisture, 1.0% fat, 29.5% protein, and 3.1% ash content.

### 3.2. Dough Fundamental Rheological Properties

The experimental data for creep compliance were well adjusted (R^2^ > 0.97) to the Burger’s model, the parameters for which are presented in Table 1. In this table, it can be observed that in general, the addition of LGF influenced all the values for the creep parameter. Compared to the control sample, the addition of LGF decreased the value of the J_Co_ parameter. At an LGF addition of 10%, the value of this parameter significantly increased (*p* < 0.05) but did not exceed that for the control sample. In general, the creep compliance (J_C0_, J_Cm_, and J_max_) values decreased with an increased addition of LGF. Only at an addition of 10% LGF in the wheat flour did the value of these parameters significantly increase (*p* < 0.05), but without exceeding the values for the control sample. In the case of the λ_C_ parameter (creep retardation time), we observed that its value was significantly increased (*p* < 0.05) by an addition of 5% LGF, then its value significantly decreased (*p* < 0.05) but was not lower than in the case of the control sample. Regarding the μ_Co_ parameter, we observed that it generally registered a higher value for the samples with LGF addition.

In the case of the recovery test we noticed that the compliance parameter was significantly influenced (*p* < 0.05) depending on the LGF level addition in the dough sample. Considering the sample without LGF addition, the value of the λ_R_ parameter was lower in the case of the addition of 5% and 10% LGF and higher in the cases of an addition level of 2.5% and 7.5% LGF. The J_r_/J^max^ ratio had higher values for the bread containing LGF.

### 3.3. Dough Microstructure

Figure 1 shows the microstructure obtained using EFLM for the dough without LGF addition (Figure 1A) and those with LGF addition (Figure 1B–E). In these images, it can be seen that as the level of LGF addition increased, the red-colored area increased. Because the red-colored area corresponded to rhodamine B, which marked the proteins in the dough matrix, it meant that the LGF addition led to an increase in the amount of protein in the dough system. Rhodamine B exhibits hydrophobic affinities for protein-rich domains [32], which color them in red. Starch granules were highlighted by the green coloration specific to fluorescein, a color that was more predominant in the sample without LGF addition in the dough recipe.

### 3.4. Bread Quality Evaluation

#### 3.4.1. Bread Physical Characteristics

The physical characteristics of the bread samples used in Table 2 are important because they influence the way consumers perceive the quality of bread. As can be seen, compared to the control sample, the specific volume of the bread samples significantly increased (*p* < 0.05) with an increasing level of LGF and reached a maximum value at 7.5% LGF incorporated in the bread recipe. At an addition level of 10% LGF, the specific volume of the bread samples decreased but was not lower than in the case of the sample without LGF addition in the bread recipe. Regarding the porosity of the samples, it can be seen in our data that the addition of LGF significantly improved (*p* < 0.05) this parameter’s value. At an addition level of 10% LGF in wheat flour, the value of the porosity began to decrease significantly (*p* < 0.05). The bread elasticity was improved due to the LGF addition even at an addition level of 7.5% LGF in the wheat flour.

#### 3.4.2. Color Parameters of Bread Samples

Table 3 shows that in the case of the crust, the value of the *L** parameter (darkness/brightness) significantly decreased (*p* < 0.05) due to the addition of LGF in the bread recipe, which meant that the samples with LGF addition had a darker crust. The same trend could seen for the bread crumb. Regarding to the *a** parameter (shade of green/red), we observed that its value was significantly increased (*p* < 0.05) due to the LGF addition, both in the case of the crust and the crumb. This meant that the LGF addition gave the bread samples a slightly reddish color. The *b** parameter (shade of blue/yellow) showed a significantly increasing value (*p* < 0.05) as the LGF addition value increased, which meant that the yellow hue of the bread samples was intensified due to the addition of LGF in the bread recipe.

#### 3.4.3. Texture Profile Analysis of Bread Samples

Table 4 shows the textural parameters (firmness, gumminess, cohesiveness, and resilience) of the bread samples with different levels of LGF addition. As can be seen in the data obtained, the firmness of the samples significantly increased (*p* < 0.05) with an increase in the level of LGF in the wheat flour. The gumminess parameter significantly increased (*p* < 0.05) at an addition level of 2.5% and 5% LGF. Then, its value decreased but did not go lower than in the case of the control sample without LGF addition. The cohesiveness of the bread samples showed a decreasing trend as the level of LGF addition increased. The resilience also decreased with an increasing level of LGF in the wheat flour.

#### 3.4.4. Crumb Structure of Bread Samples

Figure 2 shows the crumb structure of the bread samples. In these images, it can be seen that the addition of LGF in the bread samples positively influenced their crumb structures. Consumers generally prefer bread with fine pores and a uniform density. While taking into account the structure of the pores in the case of the control sample, it was observed that for the samples with an LGF addition of 2.5% and 5%, the pores of the crumb were more uniform and denser. In contrast, once the level of LGF addition was higher than 5%, the pores were less dense than in the case of the control sample.

#### 3.4.5. Sensory Analysis of Bread Samples

In general, the addition of LGF in the bread recipe was well appreciated by panelists. In Figure 3, it can be observed that the incorporation of LGF in the bread recipe changed all the sensory characteristics evaluated by the panelists. Thus, it was observed that regarding the color, the sample with 7.5% LGF addition followed by the one with 5% LGF addition were the best appreciated. In addition, the taste, smell, texture, flavor, and global acceptability had the highest scores for the samples with 7.5% and 5% LGF addition. The samples with the minimum value of the LGF addition (2.5%) and the maximum value of the LGF addition (10%) were appreciated less by the panelists.

## 4. Discussion

### 4.1. Dough Fundamental Rheological Properties

In the case of the creep test, the increase in the J_Co_ parameter could be correlated with a significant (*p* < 0.05) decrease in the instantaneous elasticity of the dough. The lowest value for the J_Co_ parameter was obtained in the case of the incorporation of 7.5% LGF in the wheat flour. An increase in the value of the λ_C_ parameter indicated the fact that the retarded elastic creep took place more slowly. A lower value of the λ_R_ parameter at an addition level of 5% and 10% LGF compared to the control sample highlighted the fact that the retarded elastic recovery took place more rapidly for these samples. A higher value for the μ_Co_ parameter for the samples with high levels of LGF addition indicated a higher resistance to flow deformability by the dough compared to the control sample. However, the sample with a 2.5% LGF addition in the wheat flour had a lower value for the μ_Co_ parameter, which meant that it presented a lower opposition to deformation. In general, the effect of the LGF addition in the wheat flour on the dough behavior could be explained by the dilution of gluten, which changed its elasticity and viscosity. The J_max_ parameter presented a decreasing trend due to the LGF addition in the dough recipes. This trend was similar to those reported in other studies that concluded that the addition of 15% and 20% roasted chickpea flour in the dough recipe had the effect of decreasing the J_max_ value [33].

The results of the recovery test highlighted the effect of LGF addition on the dough elasticity. According to our data, the control sample had the lowest elasticity while the sample with 10% LGF addition had the highest elasticity and presented the highest value for the J_Ro_ parameter. The fact that the value of the compliance elasticity during the recovery parameter was higher indicated a higher recoverable energy. This recoverable energy was stored by more cross-linked gluten than in the case of the sample without LGF addition in the dough recipe. This meant that with the LGF addition, the aggregation between gluten proteins increased. The observed increase in the values for the λ_R_ parameter for the samples with LGF addition meant that the retarded elastic recovery of dough was slower due to the LGF addition in the wheat flour. The J_r_/J_max_ ratio provided information on the elasticity of the dough samples. According to Moreira et al. [34], the increase in the value of this ratio indicated the fact that the elastic properties of the dough became more pronounced after LGF addition in the dough system.

### 4.2. Dough Microstructure

As the EFLM images show, the control sample without LGF addition had the highest amount of green area, which indicated a high starch amount. The samples with the LGF addition in the dough system were highlighted by slightly different EFLM images due to a more nuanced red coloration specific to rhodamine B that was correlated with the amount of protein in the system. Thus, the EFLM images showed a higher protein content and a lower amount of starch once the LGF addition level was increased. This was expected due to the fact that white wheat flour contains a much lower amount of protein than lentils [35]. Moreover, in a previous study we concluded that during the germination process, the amount of protein in the lentil grains increased [36]. Similar results in the sense of increasing the amount of protein and decreasing the amount of starch due to the addition of lentil flour in the dough system were previously reported [20]. As could be seen, for the control sample and for the dough samples with 2.5% and 5% LGF incorporation in the bread recipe, the starch granules were agglomerated and concentrated in different areas. As the LGF level was increased to high levels of 7.5% and 10%, the starch granules appeared to be more dispersed among the protein granules. At high amounts of the LGF addition, the starch granules appeared to be separated and surrounded by those of protein. Instead of this, according to the images obtained, all the dough samples were homogeneous in the sense that no black regions were observed in the matrix of the dough. This may have indicated that the LGF addition did not lead to a weakening of the dough. These results were also correlated with the results obtained for the bread samples that showed that even at an incorporation of 10% LGF in the wheat flour, the quality characteristics of bread were not negatively influenced compared to the control sample.

### 4.3. Bread Quality Evaluation

#### 4.3.1. Bread Physical Characteristics

The incorporation of LGF in the technological process of bread making at a level of 2.5–10% in the wheat flour influenced each physical parameter of the bread samples. The improvement in specific volume was similar to the porosity and elasticity increases. This was desirable because these parameters are indicators of bread quality. According to other studies, the use of different leguminous flours as ingredients in bread making decreased these values [37]. However, it was observed that the incorporation of germinated pulses in flour reduced this inconvenience. Moreover, in some cases this incorporation led to an improvement in the physical parameters of bread samples [38].

Regarding the improvement in the bread volume due to the addition of LGF, we can state that this was due to the changes that occurred during germination in the gelatinization of starch and in the aggregation of proteins. Thus, in the germination process there was an increase in protein solubility that coincided with the improvement in the ability of the lentil flour to foam and emulsify. The germination process led to an increase in the proteolytic activity, which led to the degradation of the lentils’ storage proteins. This led to shorter peptides and free amino acids. As a result, the solubility of the proteins increased [13]. The change in the sense of increasing the volume of bread with the addition of LGF could be attributed to the fact that during germination, there was an increase in the amount of simple, fermentable sugars. These sugars were used by yeast in the fermentation process, which resulted in the production of CO_2_ [39]. The final result was an increase in the bread volume. Similar results in the sense of increasing the volume of bread due to the addition of sprout flour were found in another previous study [40]. In addition, the increase in the bread volume due to the LGF incorporation may be explained by the increase in α-amylase activity during the germination process. The activity of α-amylase had a positive role in the process of the bread making, as was concluded in another study [41], because it had a role in the starch degradation, which favored the activity of the *Saccharomyces cerevisiae* baking yeast. Moreover, amylases also influenced the process of slowing the staling of bread [42]. Similar results in the sense that the germination process could be used to improve the volume of bread were reported in a previous study. This study found that germination positively influenced the behavior of whole wheat flour in the process of bread making [43]. According to our data, the volume of the bread samples with LGF addition was higher than that of the control sample even when high levels of LGF were incorporated in the bread recipe. These results indicated that the LGF addition improved the dough behavior during fermentation and was able to retain the gas formed in a better way than did the control sample. However, when an addition level of 10% LGF was incorporated in the bread recipe, the specific volume of bread began to decrease but was still significant higher (*p* < 0.05) than that of the control sample. The significant decrease (*p* < 0.05) in the bread sample with 10% LGF addition may have been due to a slight dough weakening as a consequence of the gluten amount decreasing in the dough system while taking into account that the LGF did not contain gluten.

The addition of LGF in the bread-making recipe had the effect of improving the porosity of the bread. However, after exceeding a 7.5% level of LGF addition, the porosity began to decrease slightly but did not fall below the value for the control sample. The significant increase (*p* < 0.05) in the bread porosity due to the LGF addition may be explained by the fact that the capacity to produce and retain gases in the dough proofing stage increased, leading to a denser porous structure. Similar results in the sense that the germination process could be successfully used to improve the porosity of the bread were reported by Cardone et al., who concluded that the germination process under controlled conditions for durum wheat grains had the effect of improving the quality characteristics of bread of volume and porosity [44]. The significant decrease (*p* < 0.05) in porosity values at high levels of LGF addition in the wheat flour may be attributed to the decrease in the amount of gluten from the dough matrix. Due to its substitution with LGF flour, there was a decrease in the amount of gluten in the dough system. This also led to the weakening of the dough network, which may have presented a lower capacity to form pores in the proofing stage and a lower capacity to retain carbon dioxide in the dough system.

Regarding the elasticity value, in this study we observed that the LGF addition led to an improvement in this parameter even at an addition level of 7.5% in the wheat flour. The significant increase (*p* < 0.05) in the elasticity value of the bread samples can be explained by the presence of the enzyme α-amylase in a larger amount when the addition level of LGF in the bread recipe was increased. These data were similar to those from another study that concluded that the amylolytic enzymes improved the physical characteristics of bread and the viscoelastic properties of wheat dough [45].

#### 4.3.2. Color Analysis of Bread Samples

The incorporation of LGF in the bread making changed all three color parameters (for the crust and for the bread crumb). The significant decrease (*p* < 0.05) in the *L** parameter value with an increased level of LGF in the wheat flour indicated a darkening of the color of the bread. This can be attributed to the fact that the LGF contained a higher amount of protein than the white wheat flour. Similar data were also reported by Xing et al., 2021, who concluded that protein addition in wheat bread had the effect of darkening the bread [46]. A darker color of bread samples with LGF addition may be attributed to the intensification of Maillard reactions due to the increased amount of amino acids and reducing sugar [47]. The darkening of the bread samples can also been explained by the fact that the addition of LGF led to an increase in the amount of phenolic compounds, which coincided with a darker color [48]. Some studies previously concluded that during germination, there was an increase in the amount of phenolic compounds from the grains [49]. This may be explained by the fact that during the germination of grains, a series of biochemical changes take place that coincide with the production of secondary metabolites (such as phenolic compounds) [49]. The value of the parameter *a** (shade of green/red) increased in the cases of the crust and of the bread crumb, which indicated that the samples acquired a slightly reddish hue. This was particularly due to the specific pigments in the red lentils used as an addition in this study. Studies have shown that red lentils contain a significant amount of carotenoids [50]. The increase in the value of parameter *b** (shade of blue/yellow) due to the LGF addition indicated that it had given a yellow hue to the bread samples. This was due to the increase in the amount of protein in the nutritional composition of the bread once the addition level of LGF had increased. Along with sugars, amino acids are precursors for Maillard reactions [51]. Borrelli and Fogliano (2005) stated that melanoidins in bread had a protein skeleton [52].

#### 4.3.3. Texture Profile Analysis of Bread Samples

The firmness parameter of the bread samples was influenced by the LGF addition in the bread recipe. Its value decreased with an increase in the level of LGF addition in the wheat flour. This can be explained by the fact that once the quantity of proteins in the bread matrix had increased, the interactions between them and the starch changed [53]. The increase in the firmness value also may have been caused by the interactions between gluten in the wheat flour and dietary fibers from the LGF. With the addition of LGF, which is high in dietary fibers [54], a decrease in the strength of the gluten network may be caused due to the higher water-binding capacity of dietary fibers [55]. A higher gumminess value for the bread samples indicated a more compact structure of the bread. The fact that the bread samples with the incorporation of LGF had a significant higher value (*p* < 0.05) for the gumminess parameter compared to the control sample may be explained by the fact that the LGF addition introduced an additional amount of fiber. A higher amount of fiber led to an increase in the gumminess of the bread, as was reported in a previous study [56]. Higher gumminess values for bread may also have been caused by the modifications that occurred in the hierarchical structure of the starch due to α-amylase that was activated during the germination process. Some studies showed that gumminess is related to short-range ordered structures, helical structures, and crystalline starch structures. Changes in the structure of starch were correlated with changes in viscosity due to the LGF incorporation in the wheat flour. The addition of LGF led to changes in the hierarchical structure of starch due to α-amylase activity, which affected the gumminess of the bread [57]. The decrease in the values for the bread parameters of cohesiveness and resilience can be explained by the dilution of the gluten due to the LGF addition due to the decrease in the strength of the gluten network [58].

#### 4.3.4. Crumb Microstructure of Bread Samples

The crumb structure of bread samples is an important sensory feature for consumers. The addition of LGF in the wheat flour modified both the structure of the pores and their density. The change in the structure of the pores could be explained by the fact that the LGF addition in the wheat flour modified the gluten matrix: the dough had a lower capacity for gas expansion during the proofing and prebaking stages. Another explanation could be the fact that a larger amount of fiber due to the LGF addition led to a decrease in the bread’s ability to retain gases in the dough matrix [59]. A denser pore distribution can be attributed to an increase in the quantity of enzymes from the dough, which had the effect of influencing the fermentation stage because the enzymes hydrolyzed the starch into fermentable sugars that were later used by yeast [60]. This led to the release of a larger amount of CO_2_ and a denser distribution of pores. An uneven pore distribution can be attributed to the substitution of gluten in the dough matrix due to LGF incorporation, which led to a lower capacity of the dough to retain the released gas [5].

#### 4.3.5. Sensory Analysis of the Bread Samples

The sensory characteristics of the bread samples with the LGF incorporation in the wheat flour were more appreciated by the panelists. The appearance of the bread was better evaluated by the panelists, probably due to the fact that LGF addition in the bread recipe improved the loaf volume and porosity. The color of the bread became darker with the increase in the level of LGF addition in the wheat flour, which led to the idea of wholemeal wheat bread correlated by the panelists with a superior nutritional value compared to the white wheat flour. The LGF addition in the wheat flour also intensified the Maillard reaction, which provided more copper-colored samples that were more pleasing visually. Regarding the taste of the samples, we can state that it was improved because, as already highlighted in other previous studies, germination had the effect of improving the taste and aroma profile of the grains subjected to this process. During germination, there was a release of the reducing sugars and amino acids from the grains. In the bread-baking stage, they reacted and formed melanoidins [61]. In addition, the activation of amylolytic enzymes during germination led to starch hydrolysis, which can impart a certain level of sweetness to the grains [62]. The texture of the bread samples with LGF addition in the bread recipe was mainly improved due to the activation of amylolytic enzymes during germination; these enzymes had a positive effect on the specific volume and porosity of the bread samples. The same results were highlighted by Montemurro et al., 2019, who concluded that the sensory profile of the bread samples could be positively influenced by the germination process [63].

## 5. Conclusions

The germinated lentil flour addition in wheat flour led to significant changes in the dough microstructure, rheological behavior, and bread quality. Depending on the percentage of LGF addition in the wheat flour, its influences on dough rheology, bread quality, and consumer acceptability were different. Creep and recovery tests showed a higher viscosity at steady state (μ_C_) and a lower elastic deformation at constant stress (J) for dough samples with LGF addition. The EFLM images showed a homogeneous, compact dough matrix for all the analyzed dough samples even when high levels of LGF were incorporated in the dough recipe. The bread quality was improved by LGF addition in the bread recipe. The specific volume, porosity, and elasticity all presented higher values for bread samples with LGF addition compared to the control sample. The color parameters indicated a darkening of the bread crumb and crust via a decrease in the *L** value and an increase in the *b** and *a** values. For all the bread samples with LGF incorporated in bread recipe, the textural parameters of firmness and gumminess presented higher values compared to the control sample, whereas the cohesiveness and resilience presented lower values. The bread crumb structure with LGF incorporated in the bread recipe presented changes, especially for the bread samples in which high levels of LGF were incorporated in the wheat flour. When considering the results of the sensory analysis, we concluded that the bread samples with LGF addition in the bread recipe were better appreciated than the control sample. When taking into account the results obtained for the dough samples with different levels of LGF addition and also those for the bread samples, a value of 7.5% is recommended as the optimal level for LGF addition in bread recipe. This LGF addition level had the best overall acceptability. According to our study, LGF proved to be a valuable functional ingredient for food producers that provide natural bakery products without other additives or enzymatic addition and of a very good quality. In addition, it may improve the nutritional quality of bakery products due to the nutrients provided by LGF.

## Figures and Tables

**Figure 1 foods-11-02982-f001:**
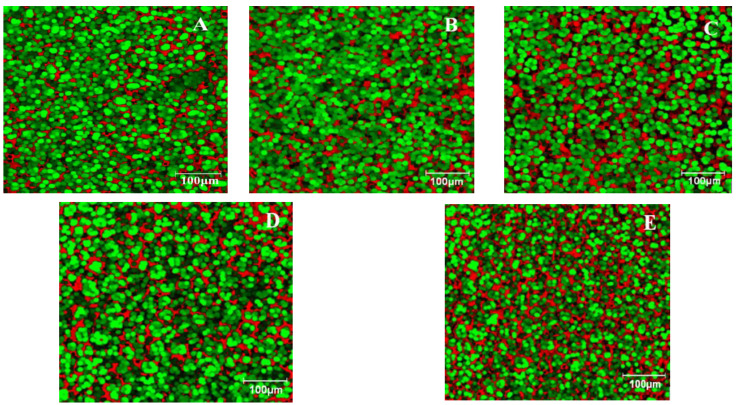
Microstructure obtained using EFLM of the wheat dough with LGF at different levels: 0% (**A**); 5% (**B**); 10% (**C**); 15% (**D**); 20% (**E**). Red, protein; green, starch granules.

**Figure 2 foods-11-02982-f002:**
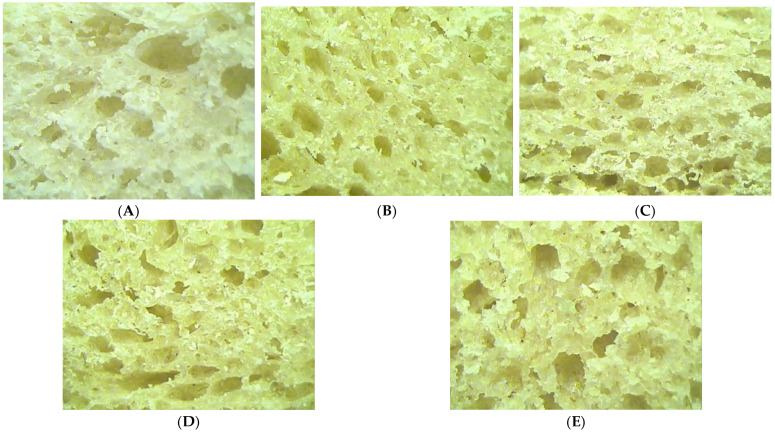
Structure of wheat dough with germinated lentil flour (LGF) at different levels: 0% (**A**); 2.5% (**B**); 5% (**C**); 7.5% (**D**); 10% (**E**).

**Figure 3 foods-11-02982-f003:**
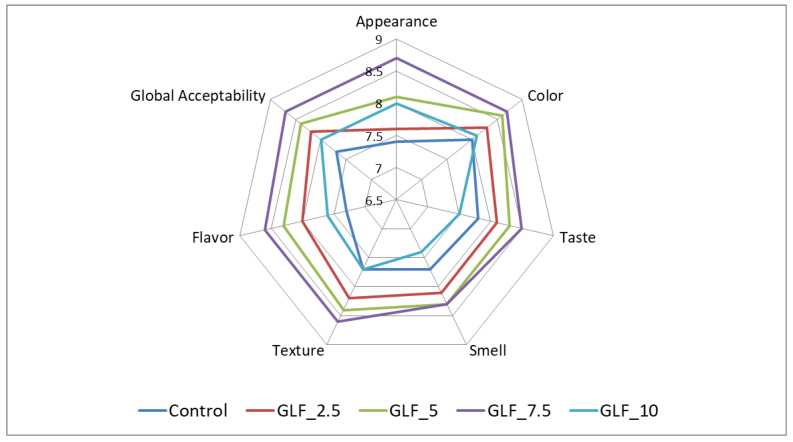
Sensory analysis of bread samples.

**Table 1 foods-11-02982-t001:** Parameters of Burger’s model.

Samples	Creep Phase					Recovery Phase				
	J_Co_—10^5^ (Pa^−1^)	J_Cm_—10^5^ (Pa^−1^)	λ_C_ (s)	μ_Co_—10^−6^ (Pa—s)	J_max_—10^5^ (Pa^−1^)	J_Ro_—10^5^ (Pa^−1^)	J_Rm_—10^5^ (Pa^−1^)	λ_R_ (s)	J_r_—10^5^ (Pa^−1^)	J_r_/J_max_ (%)
Control	6.93 ^e^ (0.02)	20.00 ^b^ (0.01)	34.99 ^a^ (0.05)	0.57 ^a^ (0.00)	24.76 ^a^ (0.02)	8.66 ^c^ (0.04)	8.32 ^d^ (0.00)	34.36 ^a^ (0.00)	16.98 ^d^ (0.04)	68.57 ^a^ (0.14)
LGF_2.5	5.86 ^c^ (0.02)	20.00 ^b^ (0.00)	35.33 ^a^ (0.01)	0.56 ^a^ (0.00)	21.78 ^c^ (0.02)	7.93 ^b^ (0.02)	7.34 ^c^ (0.00)	38.20 ^b^ (0.58)	15.27 ^c^ (0.02)	70.09 ^ab^ (0.01)
LGF_5.0	5.13 ^b^ (0.03)	20.01 ^b^ (0.01)	39.73 ^d^ (0.09)	0.64 ^b^ (0.00)	20.47 ^b^ (0.03)	8.78 ^c^ (0.03)	6.11 ^a^ (0.00)	31.94 ^a^ (0.00)	14.89 ^b^ (0.03)	72.77 ^c^ (0.04)
LGF_7.5	4.45 ^a^ (0.26)	10.00 ^a^ (0.00)	37.30 ^c^ (0.25)	0.75 ^c^ (0.02)	16.49 ^a^ (0.03)	5.29 ^a^ (0.30)	6.24 ^b^ (0.06)	40.22 ^b^ (2.01)	11.52 ^a^ (0.24)	69.90 ^ab^ (1.55)
LGF_10	6.60 ^d^ (0.02)	20.00 ^b^ (0.00)	36.45 ^b^ (0.16)	0.58 ^a^ (0.01)	24.49 ^d^ (0.03)	10.00 ^d^ (0.00)	7.35 ^c^ (0.00)	34.01 ^a^ (0.00)	17.35 ^e^ (0.00)	70.85 ^b^ (0.09)

J_Co_, instantaneous compliance of creep phases; J_Cm_, retarded elastic compliance or viscoelastic compliance of creep phases; λ_C_, retardation time; μ_Co_, zero shear viscosity; J_max_, maximum creep compliance obtained at the end of the creep test; J_Ro_, instantaneous compliance of recovery phases; J_Rm_, retarded elastic compliance or viscoelastic compliance of recovery phases; λ_R_, mean retardation time of recovery phases; J_r_, recovery compliance evaluated when dough recovery reached equilibrium; J_max_, maximum creep compliance value in the creep phase for 60 s, which corresponds to the maximum deformation. Values in parentheses are standard deviations. Means followed by the same letter within a column were not significantly different. Different letters (^a,b,c,d,e^) within the same column for each parameter indicate that the means were significantly different (*p* < 0.05).

**Table 2 foods-11-02982-t002:** Physical characteristics of the bread samples with different levels of LGF addition in wheat flour.

Bread Samples	Specific Volume (cm^3^/100 g)	Porosity (%)	Elasticity (%)
Control	331.5 ± 0.74 ^a^	67.4 ± 0.86 ^a^	91.3 ± 0.57 ^b^
LGF_2.5	351.2 ± 1.02 ^b^	72.8 ± 1.31 ^b^	93.5 ± 0.37 ^c^
LGF_5.0	366.2 ± 0.98 ^c^	78.5 ± 0.66 ^d^	94.5 ± 0.45 ^c^
LGF_7.5	375.0 ± 2.33 ^d^	79.8 ± 0.30 ^d^	95.8 ± 0.26 ^d^
LGF_10	351.1 ± 1.05 ^b^	75.4 ± 0.53 ^b^	89.8 ± 0.58 ^a^

The results are the mean ± standard deviation (*n* = 3). Bread samples containing germinated lentil flour, LGF: a–d, mean values followed by the same letter within a column were not significantly different (*p* < 0.05).

**Table 3 foods-11-02982-t003:** Color parameters of the bread samples with different levels of LGF addition in wheat flour.

Bread Samples	Crust Color			Crumb Color		
	*L**	*a**	*b**	*L*	*a**	*b**
Control	76.25 ± 0.94 ^c^	3.44 ± 0.27 ^a^	3.14 ± 0.43 ^a^	66.37 ± 0.88 ^c^	−4.62 ± 0.32 ^d^	1.69 ± 0.22 ^a^
LGF_2.5	59.08 ± 0.94 ^b^	12.15 ± 0.50 ^b^	4.35 ± 0.39 ^b^	59.01 ± 0.95 ^b^	−3.30 ± 0.16 ^c^	1.75 ± 0.29 ^a^
LGF_5.0	57.68 ± 0.50 ^b^	17.38 ± 0.08 ^c^	5.13 ± 0.04 ^b^	58.42 ± 0.53 ^b^	−2.57 ± 0.43 ^b^	2.21 ± 0.11 ^a^
LGF_7.5	53.91 ± 0.87 ^a^	18.08 ± 0.14 ^c^	6.43 ± 0.12 ^c^	54.48 ± 0.45 ^a^	−1.18 ± 0.14 ^a^	3.18 ± 0.03 ^b^
LGF_20	53.08 ± 0.62 ^a^	19.07 ± 0.22 ^d^	7.35 ± 0.37 ^d^	53.43 ± 0.50 ^a^	−0.61 ± 0.07 ^a^	4.60 ± 0.23 ^c^

The results are the mean ± standard deviation (*n* = 10). Bread samples containing germinated lentil flour, LGF: a–d, mean values followed by the same letter within a column were not significantly different (*p* < 0.05).

**Table 4 foods-11-02982-t004:** Texture parameters of the bread samples with different levels of LGF addition in wheat flour.

Bread Samples	Firmness (*N*)	Gumminess (*N*)	Cohesiveness (Adimensional)	Resilience (Adimensional)
Control	9.01 ± 3.06 ^a^	7.23 ± 1.73 ^a^	0.82 ± 0.03 ^c^	1.72 ± 0.04 ^c^
LGF_2.5	12.60 ± 0.60 ^a^	9.31 ± 0.43 ^ab^	0.55 ± 0.13 ^b^	0.95 ± 0.09 ^b^
LGF_5.0	14.63 ± 0.67 ^bc^	9.43 ± 0.11 ^b^	0.47 ± 0.07 ^ab^	0.89 ± 0.08 ^b^
LGF_7.5	17.25 ± 0.44 ^c^	8.24 ± 0.11 ^ab^	0.38 ± 0.04 ^ab^	0.77 ± 0.15 ^ab^
LGF_10	18.21 ± 0.17 ^c^	8.07 ± 0.08 ^ab^	0.34 ± 0.05 ^a^	0.54 ± 0.06 ^a^

The results are the mean ± standard deviation (*n* = 3). Bread samples containing germinated lentil flour, LGF: a–c, mean values followed by the same letter within a column were not significantly different (*p* < 0.05).

## Data Availability

The datasets generated for this study are available on request to the corresponding author.

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
