# Peer review of "Effects of Germinated Lentil Flour on Dough Rheological Behavior and Bread Quality"

_foods, 2022, doi:10.3390/foods11192982_

Round 1

Reviewer 1 Report

This work studies the influence of different adding levels of Germinated lentil flour on dough rheological behavior and bread quality. It contains interesting information but revision is required. There are many writing and formatting issues to be corrected. It’s strongly suggested that the whole manuscript should be further checked by English language editing services.

Some comments are suggested in the attachment.

Author Response

30 August 2022

Dear Referee,  

We would like to thank the referee for the close reading and for the proper suggestions. We hope that we provide all the answers to the reviewer’s comments.

Thank you very much for the recommendations to publish our paper entitled “Effects of Germinated Lentil Flour on Dough Rheological Behavior and Bread Quality”.

The present version of the paper has been revised according to the reviewer’s suggestions.             

We uploaded the corrected version of the article for which we used the yellow color for the addition text.

GENERAL COMMENTS:

Referee comments: This work studies the influence of different adding levels of Germinated lentil flour on dough rheological behavior and bread quality. It contains interesting information but revision is required. There are many writing and formatting issues to be corrected. It’s strongly suggested that the whole manuscript should be further checked by English language editing services.

Response: We would like to thank to the referee for his/her close reading of our manuscript data. The whole manuscript has been revised by an English teacher.

Referee comments: Some comments are suggested:

  1. Line 22: “LGF addition had denser pore and smaller.”, smaller ??, a word maybe missing here.

Response: We have checked and corrected the sentence.

  1. Introduction Section: Advance on the effect of germination on the nutritional quality of lentil should be briefly emphasized in this section.

Response: In the introduction section, we have now briefly emphasized/completed the advantages of germination on the nutritional quality of lentil.

  1. Line 97-99: This sentence is confused, please modify it.

Response: We have modified the sentence from lines 97-99.

  1. Line 100-101: The parameter of “germination time” should be added here.

Response: We have added the germination time term in these lines according to the referee suggestions.

  1. Line 117-118: It is better to modify this sentence to “For these tests, a non-serrated parallel plate geometry with a diameter of 40mm and a gap width of 2 mm was used.”

Response: We have made the modification suggested.

  1. Line 134-137: The statement of the proportion of ingredients is not clear. Whether the ratio of GLF (2.5%, 5%, …) is taken into account of the added water? If it is, please list the all the adding proportions of ingredients (preferably in table form). If it is not, please describing the added water in another sentence.

Response: We completed in the manuscript the water addition levels used according to the water absorption capacity value of wheat-germinated lentil flours.

  1. Line 151-153: It is better to add cited references here.

Response: We have added two bibliographic sources that present the methods we used.

  1. Line 162: The word of “highlight” was used many times in this manuscript. It can be replaced by “analyze”, or “determine”, or “assay” or “measure” in the section of “Materials and Methods”.

Response: We have replaced the word "highlight" with an appropriate synonym, depending on the context according to the referee suggestions.

  1. Line 170: It is better move to the first sentence to the section of “4. Discussion” or delete it.

Response: We have moved the first sentence in the "4.Discussion" section.

  1. Results Section: For the comparison analyses of the results, whether statistically significant or not should be added in the text.

Response: We added the statistical significant in the text when we compared the results .

  1. Line 187 & 191: Please replace of “fallowing” with “following”.

Response: We have replaced “fallowing with “following” according to the referee suggestions..

  1. Line 191& 197 & 198: Please replace of “Germinated lentil flour” or “germinated lentil flour” with “LGF”.

Response: We have replaced “Germinated lentil flour” and “germinated lentil flour” with “LGF” according to the referee suggestions.

  1. Line 195: A symbol of space is missing between “R2” and “>”.

Response: We have inserted the symbol of space that was missing between “R2” and “>”according to the referee suggestions..

  1. Line 196: It is better to replace of “highlighted” with “presented”.

Response: We have replaced “highlighted” with “presented” according to the referee suggestions..

  1. Line 201: Please explain the meaning of “with the increase of the of”.

Response: Sorry, there was a mistake. We have eliminated the repetition: „of the”.

  1. Line 211: Please replace of “an addition” with “the addition”.

Response: We have replaced “an addition” with “the addition”.

  1. Line 223: It is better to replace of “Figure 1 highlights the microstructure taken by EFLM” with “Figure 1 shows the microstructure measured by EFLM”.

Response: We have replace “Figure 1 highlights the microstructure taken by EFLM” with “Figure 1 shows the microstructure measured by EFLM” according to the referee suggestions.

  1. Line 233: Please replace of “Figure 2” with “Figure 1”.

Response: We want to thank to the referee for the close reading of our manuscript. We have replaced “Figure 2” with “Figure 1” according to the referee suggestions.

  1. Line 240-242: Rewrite this sentence in a clearer way. I suggest that it can be modified to “Compared to the control sample, the specific volume of the bread samples increased with the adding amount of LGF and reached a maximum at 7.5%”.

Response: We have rewritten this sentence, according to the referee suggestions.

  1. Line 253: Please replace of “shown” with “shows”.

Response: We have replaced “shown” with “shows” according to the referee suggestions.

  1. Line 254 & 268: “decreased more and more” and “increased more and more” are poorly written. Please modify them.

Response: We have rewritten and eliminated the wording "more and more" according to the referee suggestions.

  1. Line 282-284: The sentence of “Compared to the control sample, … and they had a higher density.” is poorly written. Please modify it.

Response: We have modified the sentence from line 282-284 to be more understandable.

  1. Line 326-328: The sentence of “The fact that the value of …. than the sample without LGF addition in dough recipe.” is confused. I suggest splitting it into two sentences.

Response: We have split the sentence and we have modified it to be more understandable.

  1. Line 347: Please, explain the criteria to consider low level or high level of LGF addition.

Response: We have explained what low level and high level of addition mean.

  1. Line 337-356 the section of “Dough microstructure”: If possible, the results should be compared to the similar data presented in previous studies and fully discussed.

Response: We completed the paragraph and compared our data with similar ones from previous studies according to the referee suggestions.

  1. Line 388-391: The last sentence of this paragraph is too long and confused. I suggest splitting it into two or three sentences.

Response: We have split the sentence and also we have rewritten the sentences in order to be more understandable.

  1. Line 395: Please delete the words of “, without addition”.

Response: We have deleted the words “, without addition” according to the referee suggestions.

  1. Line 399: Please replace of “which” with “who”.

Response: We have replaced the word “which” with “who” according to the referee suggestions.

  1. Line 411: Please delete the word of “had”. Please replace of “shown” with “showed”.

Response: We have deleted the word “had” and we have replaced the word “shown” with “showed” according to the referee suggestions.

  1. Line 412: Please replace of “improves” with “improved”.

Response: We have replaced the word “improves” with “improved” according to the referee suggestions.

  1. Line 417: Please replace of “increase” with “increasing”.

Response: We have replaced the word “increase” with “increasing” according to the referee suggestions.

  1. Line 425: The statement of “which according to Özcan, 2022 lead to a darker color [41].” is poorly written. I suggest modifying it to “which lead to a darker color [41].”

Response: We have modified the statement, in accordance with the referee suggestions.

  1. Line 438: “Borrelli et Fogliano (2005)”, explain the meaning of " et". Please delete “in a previous study”.

Response: We have replaced the word "et" with the word "and" and we have deleted “in a previous study” according to the referee suggestions.

  1. Line 460: Please check “with affects …”? Or maybe “which affects …”.

Response: We thank to the referee for the close reading of our manuscript. We have checked and corrected it. It was correct the wording “which affects …” instead of “with affects …”.

  1. Discussion section: The gluten or gluten network was mentioned several times in this section. Indeed, gluten network is an important factor on the bread quality. Therefore, gluten structure analysis (e.g. free sulfhydryl content, ATR-FTIR, SEM) is suggested to be performed.

Response: We want to thank to the referee for his/her suggestions. Indeed the gluten network is an important factor on the bread quality and it amount decreased due to the fact that LGF does not contain gluten. However we do not have now the possibility to make this type of analysis due to the fact that in our research laboratories there are now a series of constructions (ventilation is installed) that will last until December. But we have in our manuscript the EFLM images which indicates to a certain extent the structure of the dough samples with different levels of LGF addition.

  1. Conclusions section: Given the objective of this work mentioned in line 92-94 “Thus, highlighting the optimal level of LGF addition, …”, please indicate in the Conclusions what would be the best recipe to prepare bread with improved quality and the slightest organoleptic changes that could be accepted by the consumers.

Response: In the Conclusions section, we have completed now the optimal level of LGF addition, taking in consideration the values of the determinations made for dough and bread samples. Also we mentioned what are the slightest organoleptic changes that could be accepted by the consumers according to the referee suggestions.

Finally, the authors would like to thank reviewers for their appreciations and for all the suggestions because these helped us to correct our paper and to optimize it.

Sincerely,

Georgiana Codină et al.

Reviewer 2 Report

Manuscript ID: Foods_ 1892573

In the article entitled: “Effects of Germinated Lentil Flour on Dough Rheological Behavior and Bread Qualityauthors examine the influence of different levels of the additive of germinating lentils in wheat flour on dough rheological behaviour, dough microstructure and bread quality.

This is an interesting article. Written by people with a very high level of knowledge of the topic. This article has been written in a compact manner. From the methodological point of view, the employed measurement techniques are appropriate to the adopted objective of the research work. The results obtained may have practical application.

Title

The title and the aim of the study are clearly constructed.

Abstract

The abstract includes the aim of the study, methods used in the experiment and contain the principal results and conclusions.

Introduction

The introduction describes the matter of the experiment and states the problem being investigated. Authors correctly interpreted and described the significance of the results for the research. They skillfully referred to the results of other researchers. Literature references are the most current (usually from the last 2-3 years).

Methods

The data is well collected. The methods as far as possible described in detail. The sampling is appropriate and adequately described. However, I have a few minor comments.

2.3. Dough microstructure

Although the authors refer to the methodology described in an earlier article, in my opinion more details should be given.

2.4. Bread making

What was the humidity in the fermentation chamber and the convection oven

2.5.2. Color Parameters

What was the angle of incidence? Did the calibration of the appliance take into consideration the scattered light and white reference standard?

Results and Discussion

Authors correctly interpreted and described the significance of the results for the research.

Conclusion

The authors correctly indicate, how the results are related to the studies.

References

Literature references are appropriate and relate to the position from the last few years.

Language

The article is correctly written. English language and style are minor spell check required.

Author Response

31 August 2022

Dear Referee,  

We would like to thank the referee for the close reading and for the proper suggestions. We hope that we provide all the answers to the reviewer’s comments.

Thank you very much for the recommendations to publish our paper entitled “Effects of Germinated Lentil Flour on Dough Rheological Behavior and Bread Quality”.

The present version of the paper has been revised according to the reviewer’s suggestions.             

We uploaded the corrected version of the article for which we used the yellow color for the addition text.

GENERAL COMMENTS:

Referee comments: In the article entitled: “Effects of Germinated Lentil Flour on Dough Rheological Behavior and Bread Quality” authors examine the influence of different levels of the additive of germinating lentils in wheat flour on dough rheological behaviour, dough microstructure and bread quality.

This is an interesting article. Written by people with a very high level of knowledge of the topic. This article has been written in a compact manner. From the methodological point of view, the employed measurement techniques are appropriate to the adopted objective of the research work. The results obtained may have practical application.

Response: We would like to thank to the referee for his/her close reading of our manuscript data and his/her appreciation.

Referee comments:

Title

The title and the aim of the study are clearly constructed.

Abstract

The abstract includes the aim of the study, methods used in the experiment and contain the principal results and conclusions.

Introduction

The introduction describes the matter of the experiment and states the problem being investigated. Authors correctly interpreted and described the significance of the results for the research. They skillfully referred to the results of other researchers. Literature references are the most current (usually from the last 2-3 years).

Methods

The data is well collected. The methods as far as possible described in detail. The sampling is appropriate and adequately described. However, I have a few minor comments.

Response: We want to thank to the referee for his/her appreciation.

Referee comments:

2.3. Dough microstructure

Although the authors refer to the methodology described in an earlier article, in my opinion more details should be given.

Response: We completed now our manuscript with more details related to the microstructure methodology.

2.4. Bread making

What was the humidity in the fermentation chamber and the convection oven.

Response: We completed now in the manuscript the humidity in the fermentation chamber and the convection oven.

2.5.2. Color Parameters

What was the angle of incidence? Did the calibration of the appliance take into consideration the scattered light and white reference standard?

Response: We completed now in our manuscript the calibrated data.

Results and Discussion

Authors correctly interpreted and described the significance of the results for the research.

Conclusion

The authors correctly indicate, how the results are related to the studies.

References

Literature references are appropriate and relate to the position from the last few years.

Language

The article is correctly written. English language and style are minor spell check required.

Response: We want to thank to the referee for his/her appreciation. We have made the corrections for English language, style and spell that we thought were necessary.

Finally, the authors would like to thank reviewers for their appreciations and for all the suggestions because these helped us to correct our paper and to imrpove it.

Sincerely,

Georgiana Codină et al.

Reviewer 3 Report

Comments for Authors

I reviewed the research paper entitled “Effects of Germinated Lentil Flour on Dough Rheological Behavior and Bread Quality”. The manuscript is about the application of germinated lentil flour on dough rheology and bread quality. In my opinion, the topic taken up by the authors is not new, and the current manuscript has high similarity index >34%.

      Abstract: lines 13-16 – “Dough microstructure……” should be improved/rephrase for clarity.

      Line 21-22, “The bread samples with 2.5% and 5% LGF addition had denser pore and smaller” changed with “The bread samples with 2.5% and 5% LGF addition had smaller pores and dense structure.

      Line 22; I suggest to replace “Sensorial” by “In sensory.

      Overall, the abstract must be concise and only shows the major findings.

      Introduction: This section must be reorganized. Some parts must be reduced so more focus can be given to the germinated lentil. Please mention previous studies regarding sprouting/germinating and their benefits or its contribution to bakery products.

      Hernandez-Aguilar et al. [22] have been reported that LGF addition in bread making will conduct to a decrease of bread cohesiveness and to an increase of it hardness.

      Material and methods: storage condition and packaging material for germinated lentils flour should be added

      Method of wet gluten content should be added with reference/method number.

      I suggest to add International Association for Cereal Chemistry. ICC Standard No 104–106, 110, 136 “ICC standard methods” reference in Bibliography.

       loaf characteristics (specific volume) methods detail should be added with AACC method No……

      seed displacement method??? Detail should be provided with reference method. Please, revise it. 

      Line 189-190, According to these data, the white wheat flour was one of strong quality for bread making and had a low α-amylase activity. Add α-amylase activity result for justifying the statement???

      Line 204, “It was observed…..” comma should be added before it.

      Line 209, “significantly influenced different,” add significant level.

      Line 242, The mentioned statement “At an addition of 10% the specific volume of the bread samples decreased.” showed decreasing trend. However, Table-2 showed increasing trend with the addition of germinated lentils flour at different levels. Give specific reasoning

      In sensory assessment, describe scoring detail of each treatment.

      Line 374-381, justifying the increasing trend in specific volume… In current results, why specific volume reduces at 10% incorporation of LGF.

      Line 401-404, “The decreased in porosity….” Should be rephrased for better understanding.

      Line 445-446, dietary fiber discussed for justifying the reasoning but the dietary fiber profile of LGF was not explored in current research

      Line 468, gas expansion during the proofing and baking stages. Baking stages should be changed with pre-baking stage.

      Line 492-493, specific products of the Maillard reaction???

      Conclusions must be more focused and targeted on the overall outcome of the study. The present section is a summary of the results.

      Plagiarism should be remove as per journal instructions.

Author Response

31 August 2022

Dear Referee,  

We would like to thank the referee for the close reading and for the proper suggestions. We hope that we provide all the answers to the reviewer’s comments.

Thank you very much for the recommendations to publish our paper entitled “Effects of Germinated Lentil Flour on Dough Rheological Behavior and Bread Quality”.

The present version of the paper has been revised according to the reviewer’s suggestions.             

We uploaded the corrected version of the article for which we used the yellow color for the addition text.

GENERAL COMMENTS:

Referee comments: I reviewed the research paper entitled “Effects of Germinated Lentil Flour on Dough Rheological Behavior and Bread Quality”. The manuscript is about the application of germinated lentil flour on dough rheology and bread quality. In my opinion, the topic taken up by the authors is not new, and the current manuscript has high similarity index >34%.

Response: To our knowledge, there are no such extensive studies in the specialized literature on the effect of the addition of germinated lentil flour on the rheology of white wheat flour dough, the quality of the bread and the level of consumer acceptability. We have previously discussed the effect of adding flours from other germinated legumes (beans, lupine, chickpeas) on which effect on bread making have been published and much of the similarity comes from these already published articles. However, this article highlights the effect of adding another legume (red lentils) in bread making and these data are the most interesting ones from all our research (part of the PN-III-P1-1.1-TE-2019-0892 project) in our opinion due to the fact that for a bread of a good quality a very few amount of germinated lentil are needed (we was surprised by this fact) which may be useful and practical for the producers.

Referee comments: Abstract: lines 13-16 – “Dough microstructure……” should be improved/rephrase for clarity.

Response: We have rephrase the sentence, for more clarity according to the referee suggestions.

Referee comments: Line 21-22, “The bread samples with 2.5% and 5% LGF addition had denser pore and smaller” changed with “The bread samples with 2.5% and 5% LGF addition had smaller pores and dense structure.”

Response: We changed the sentence as we received the recommendation.

Referee comments: Line 22; I suggest to replace “Sensorial” by “In sensory”.

Response: We have replaced the word, according to the referee suggestions.

Referee comments: Overall, the abstract must be concise and only shows the major findings.

Response: We have made certain changes, so that the abstract to be more concise and to shows only the major findings.

Referee comments: Introduction: This section must be reorganized. Some parts must be reduced so more focus can be given to the germinated lentil. Please mention previous studies regarding sprouting/germinating and their benefits or its contribution to bakery products.

Hernandez-Aguilar et al. [22] have been reported that LGF addition in bread making will conduct to a decrease of bread cohesiveness and to an increase of it hardness

Response: We have reduced some parts of the introduction and we completed the introduction with more informtions on the germinated lentils (advantages of the germination process on the nutritional characteristics). Also, we have pointed out the benefits/contribution of sprouting/germination for bakery products, referring to a previous study. More studies had been added related including the mentioned one by the referee to sprouting/germinating and their benefits or its contribution to bakery products according to the referee suggestions.

Referee comments: Material and methods: storage condition and packaging material for germinated lentils flour should be added

Response: We completed the storage condition and packaging material for germinated lentils flour in the materials and methods section according to the referee suggestions. 

Referee comments: Method of wet gluten content should be added with reference/method number.

Response: We added the reference according to the referee suggestions.

Referee comments:  I suggest to add International Association for Cereal Chemistry. ICC Standard No 104–106, 110, 136 “ICC standard methods” reference in Bibliography.

Response: We have added the suggested reference in Bibliography according to the referee suggestions.

 Referee comments:     loaf characteristics (specific volume) methods detail should be added with AACC method No……

      seed displacement method??? Detail should be provided with reference method. Please, revise it. 

Response: We have presented details about seed displacement method and we have provided the reference for the method according to the referee suggestions.

    Referee comments:    Line 189-190, According to these data, the white wheat flour was one of strong quality for bread making and had a low α-amylase activity. Add α-amylase activity result for justifying the statement???

Response: We have already specified the fact that the value of the falling number index was 356 (this value is an index for α-amylase activity of the flour).     

Referee comments:    Line 204, “It was observed…..” comma should be added before it.

Response: We have placed the comma before "it was observed...." according to the referee suggestions.

Referee comments:    Line 209, “significantly influenced different,” add significant level.

Response: We have now added the significant level according to the referee suggestions.

Referee comments:    Line 242, The mentioned statement “At an addition of 10% the specific volume of the bread samples decreased.” showed decreasing trend. However, Table-2 showed increasing trend with the addition of germinated lentils flour at different levels. Give specific reasoning

Response: In the discussion section, we have explained that „at an addition of 10% the specific volume of the bread samples began to decrease, but without being lower than that of the control sample, indicates that the addition of LGF led to a weakening of the gluten network because it decreased the amount of gluten from the matrix because germinated lentil flour does not contain gluten.” Our affirmanations are in agreement with those from the table 2.

Referee comments:     In sensory assessment, describe scoring detail of each treatment.

Response: We completed now the scoring detail of each treatment in the sensory assessment.

Referee comments:    Line 374-381, justifying the increasing trend in specific volume… In current results, why specific volume reduces at 10% incorporation of LGF.

Response: In the discussion section, we have explained the reason behind the fact that at 10% incorporation of LGF the specific volume decreased.

Referee comments:    Line 401-404, “The decreased in porosity….” Should be rephrased for better understanding.

Response: We rephrased for a better understanding.

Referee comments:    Line 445-446, dietary fiber discussed for justifying the reasoning but the dietary fiber profile of LGF was not explored in current research

Response:    We have made this explanation based on studies in the field that have been carried out previously.

Referee comments:    Line 468, gas expansion during the proofing and baking stages. Baking stages should be changed with pre-baking stage.

Response:    We have changed „baking stages” with „pre-baking stage” according to the referee suggestions.

Referee comments:   Line 492-493, specific products of the Maillard reaction???

Response:    We have rephrases for a better understanding.

Referee comments:    Conclusions must be more focused and targeted on the overall outcome of the study. The present section is a summary of the results.

Response:    We completed the conclusions section according to the referee suggestions.

Referee comments: Plagiarism should be remove as per journal instructions.

Response:    We tried to reduce the plagiarism according to the journal instructions. We hope that we succeed.

We want to thank the reviewer for all the comments and his/her recommendations. We believe that help us to improve the quality of our manuscript.

Sincerely,

Georgiana Codina

Round 2

Reviewer 1 Report

no

Author Response

Thank you very much for your appreciation

Reviewer 3 Report

Now Satisfactory

Author Response

Thank you very much for your appreciation.